# A Proxy Approach to Family Involvement and Neurocognitive Function in First Episode of Non-Affective Psychosis: Sex-Related Differences

**DOI:** 10.3390/healthcare11131902

**Published:** 2023-06-30

**Authors:** Marina Soler-Andrés, Alexandre Díaz-Pons, Víctor Ortiz-García de la Foz, Nancy Murillo-García, Sara Barrio-Martínez, Margarita Miguel-Corredera, Angel Yorca-Ruiz, Rebeca Magdaleno Herrero, Jorge Moya-Higueras, Esther Setién-Suero, Rosa Ayesa-Arriola

**Affiliations:** 1Mental Illness Research Department, Valdecilla Biomedical Research Institute, 39011 Santander, Spain; 2Faculty of Psychology, University of Oviedo, 33003 Oviedo, Spain; 3Faculty of Psychology, European University of the Atlantic, 39011 Santander, Spain; 4Faculty of Psychology, National University of Distance Education (UNED), 28015 Madrid, Spain; 5Biomedical Research Center in Mental Health Network (CIBERSAM), Health Institute Carlos III, 28029 Madrid, Spain; 6Department of Molecular Biology, Faculty of Medicine, University of Cantabria, 39011 Santander, Spain; 7Faculty of Psychology, Complutense University of Madrid, 28223 Madrid, Spain; 8Department of Psychology, University of Lleida, 25001 Lleida, Spain

**Keywords:** schizophrenia spectrum disorders, first episode psychosis, family support, family involvement, sex differences, neurocognition, family centered care, women, gender roles

## Abstract

Schizophrenia spectrum disorders (SSD) often show cognitive deficits (CD) impacting daily life. Family support has been shown to be protective against CD, yet the relationship between these in psychotic patients remains complex and not fully understood. This study investigated the association between a subdomain of family support, namely, family involvement (estimated through a proxy measure), cognitive functioning, and sex in first-episode psychosis (FEP) patients. The sample included 308 patients enrolled in the Program for Early Phases of Psychosis (PAFIP), divided into 4 groups based on their estimated family involvement (eFI) level and sex, and compared on various variables. Women presented lower rates of eFI than men (37.1% and 48.8%). Higher eFI was associated with better cognitive functioning, particularly in verbal memory. This association was stronger in women. The findings suggest that eFI may be an important factor in FEP patients’ cognitive functioning. This highlights the importance of including families in treatment plans for psychotic patients to prevent CD. Further research is needed to better understand the complex interplay between family support, sex, and cognitive functioning in psychotic patients and develop effective interventions that target these factors.

## 1. Introduction

The concept of psychosis has been evolving over the last 170 years with the conceptual advances in mental disorders [1,2,3]. Despite the lack of a unified definition for the term, it is known that psychosis is a clinical syndrome rather than a nosological entity. The symptomatic presentation of a psychotic episode is identified by a range of manifestations that encompasses the presence of positive and negative symptoms, either co-occurring or occurring independently [3,4]. The former entails the presence of delusions and hallucinations, along with aberrant motor behaviors and disorganized behavior. These symptoms contribute to the distortion of perception and reality experienced by individuals during a psychotic episode. The latter involves a reduction or loss in the ability to experience pleasure, emotion, motivation, and social communication. These symptoms can significantly impact an individual’s daily functioning and overall well-being. While the ultimate causes are not yet fully understood, ongoing research within the scientific community continues to shed light on this complex matter. The psychotic spectrum continuum is currently the most accepted approach to the study of schizophrenia spectrum disorders (SSD) [4].

Along with the aforementioned symptomatology, cognitive deficit (CD) is another major feature of SSD [5] and affects nearly 98% of individuals with these disorders [6], often preceding the onset of psychotic symptoms. It is characterized by problems in various domains of cognitive functioning, including attention, learning, memory, problem-solving, and social cognition, and is associated with poorer quality of life [5]. While there is significant heterogeneity in the specific cognitive profiles of individuals with schizophrenia, researchers have identified three main areas of impairment: neurocognition, social cognition, and metacognition [7]. Moreover, studies examining sex differences in cognitive functioning among individuals with SSD have revealed that women tend to perform better on verbal tasks while men excel at visuospatial tasks [8]. Results for social cognition remain inconsistent [9].

In addition to CD, further sex differences have been broadly reported regarding other psychosis-related factors such as the age of onset [10], psychosocial stressors [11], clinical presentation [12], and caregiving responsibilities [13] among others. Research has shown that men are subject to more consistent and severe criticism from their families, which in turn increases the likelihood of relapses and unfavorable disease outcomes. In comparison, female relatives tend to exhibit greater compliance with treatment, such as attending therapies, as supported by previous studies [14]. This contrasts with men who display less tolerance for their relatives’ symptoms and feelings of responsibility [15].

The provision of family support is considered to be crucial for individuals with severe psychiatric disorders who are receiving outpatient care services [16,17]. Leal-Soto et al. (2012) [18] describe family support as: (a) Basic care: assistance related to the satisfaction of basic needs, such as food, supervision of hygiene, and adequate clothing, among others; (b) Involvement in treatment: accompanying the visits with the specialist, whether a psychiatrist, social worker, occupational therapist, and/or psychotherapy sessions; (c) Concern for medication: supervision for the proper self-administration of medications and rigor in the doses of psychotropic drugs, regarding the schedule and quantity; (d) Social interaction: integration of the patient into the social relationships of their family and peers; (e) Search for patient autonomy: the family must be able to promote the patient’s autonomy, according to their limitations and capacities; (f) Communication and expression of affection: family communication should be open and fluid, in which each member feels safe and with the opportunity to be integrated into the family conversation. Regarding the expression of affection, importance is given to the demonstration of support, recognition, and understanding.

Adopting the aforementioned framework, it is established that family involvement (FI) stands as an essential component or subdomain of family support and is, therefore, key in the care and treatment of individuals with SSD [19]. Family active participation has been shown to improve the course of the disorder, reduce relapse rates and increase treatment adherence [16,20,21]. Family interventions, such as psychoeducation, communication training, and problem-solving, have been found to be effective in reducing stress, improving coping strategies, and promoting recovery in both patients and families [21,22]. However, despite the recognized importance of family support for persons with SSD, family illness engagement rates remain poor [16,17], and have been linked to caregiving burden [23]. Insufficient awareness regarding the ailment, seclusion in therapy, and amplified obligations and financial liabilities can culminate in corporeal and cognitive onus for family members, thereby potentially prompting detachment. Previous findings have shown that a higher objective burden resulted in less displayed support from relatives [18].

Unlike other works, the present study seeks to explore, through a non-self-reported measure, a component of family support such as FI: defined as an interest in the illness and involvement in understanding it. For this purpose, participation or non-participation in the PAFIP-FAMILIAS study on relatives of patients with the first episode of psychosis (FEP) was used as a proxy measure.

Therefore, the primary objective is to explore the potential association between estimated levels of FI (eFI) and neurocognitive function in patients with first episode of non-affective psychosis, while exploring any sex-based differences in this relationship. To minimize the reliance on self-report measures, a proxy measure approach will be utilized to estimate levels of FI.

Refinement procedures have been implemented during the participant selection process to minimize potential biases and uncertainties inherent in the measure. The detailed description of these procedures is described in Section 2 and illustrated in Figure 1. Considering the scrupulous selection of participants and the subsequent measures undertaken to address the inherent limitations, it was deemed appropriate to proceed with the exploration of this measure within the scope of this study.

## 2. Materials and Methods

### 2.1. Settings and Sample

This study comprises four groups of patients with FEP who were recruited from PAFIP, a large epidemiological program for early psychosis stages conducted at the University Hospital Marqués de Valdecilla in Cantabria, Spain, from 2001 to 2018 [24,25,26,27,28,29,30,31]. In addition to its epidemiological nature, PAFIP served as an intervention program for both inpatients and outpatients with FEP who received comprehensive multidisciplinary treatment from psychiatric nursing, psychiatry, psychology, and social work during a 3-year follow-up period. FEP patients were referred from the inpatient unit, outreach mental health services, and healthcare centers in the Cantabria region. As PAFIP was the only specialized mental healthcare service for FEP in Cantabria during that period, its participants can be considered an epidemiological sample of the population in this community.

Among the 668 patients with FEP who were enrolled in PAFIP, a total of 387 fulfilled the prerequisites to participate in a family based study known as PAFIP-FAMILIAS (FIS PI17/00221) (approval numbers NCT0235832 and 2017.247) (more information in Murillo-García et al., 2022 [24]). Due to the neurocognitive and genetic nature of this study, those who did not meet the requirements were initially excluded: completing the baseline cognitive evaluation and providing DNA samples with informed consent for research purposes.

The remaining patients’ first-degree relatives were then a priori eligible to participate in this second study. Between January 2018 and March 2021, the parents and siblings of these patients were contacted by phone and invited to undergo the same neuropsychological assessment as the probands for the purposes of this second project.

Subsequently, for the present study, a final sample of 308 patients with FEP was considered according to their first-degree relatives’ explicit decision to participate or refuse participation in the PAFIP-FAMILIAS study (the remaining 79 patients were excluded due to other criteria; lack of family; being untraceable; previous indication of not to be further contacted; facing difficulties in participating).

It is fundamental to acknowledge that proxy measures, by their nature, involve certain degree of imprecision, which can affect accuracy of the measure. To increase the internal validity of the measure, it is, therefore, advisable to reduce the risk of Type I error. In order to refine the conceptual classification criteria for the study’s subjects, and to align them as closely as possible with the goal of acquiring comprehensive insight into their family member’s illness while contributing to the existing knowledge in the field (thus, suggesting a stronger commitment to their family member’s mental health), participants who, due to difficulties, were unable to accommodate convenient scheduling options for the PAFIP-FAMILIAS study (*n* = 38), were excluded from the analyzed sample.

This decision was made due to our inability to verify the genuineness of the reported difficulties in participation, which primarily involved work shifts, transportation, or health-related issues. However, it is important to note that adequate provisions and facilities were provided to help address these concerns. The aim of this exclusion was to reduce the risk of falsely identifying relatives who were not involved in their loved one’s care and to ensure to a higher degree that the included subjects truly represented the targeted population.

To support this decision, it is worth noting that study participants were afforded flexible scheduling options to mitigate any possible inconvenience associated with study participation, which is particularly important in the context of caregiving. Furthermore, the study had a mean duration of no more than 2 h and provided compensation of EUR 50 for travel and parking expenses during the evaluation. Moreover, it is important to take into account that the study was conducted at a center equipped with ample parking facilities in close proximity and convenient access to abundant public transportation connections, including buses and trains.

Therefore, considering the aforementioned, despite there may be valid reasons for declining participation which do not necessarily reflect a lack of interest in their familiar’s care, this procedure is believed to help mitigate this casuistry, remaining only those participants whose families agree to participate and those whose families decline participation due to more conviction-related reasons. Additionally, in the same step, subjects who could not be located despite repeated attempts to contact them (*n* = 19), or who were conceptually or legally incompatible with the study’s aim of examining the impact/correlation of family interest, were excluded. This comprises, respectively, patients who lacked family members (*n* = 14) or those who requested not to be further contacted for more studies in order to respect their previous wishes (*n* = 8).

In view of the facilities extended and the subsequent exclusion of subjects who, despite such measures, could not align their personal circumstances with study participation, the selection variable can be construed as reflecting voluntary and uncoerced participation. This, in turn, enhances the representativeness of the sample in relation to the target population of individuals who are involved and non-involved in caregiving.

Consequently, the study groups included PAFIP patients who were sorted out based on the level of eFI (subjects whose families participated or declined participation) and sex. The following comparison groups were created: FEP men whose families refused to participate in PAFIP FAMILIAS study (hereafter referred to as men with non-participant families) (N = 86), FEP women whose families refused to participate in PAFIP FAMILIAS study (women with non-participant families) (N = 88), FEP men whose families agreed to participate in PAFIP FAMILIAS study (men with participant families) (N = 82), and FEP women whose families agreed to participate PAFIP FAMILIAS study (women with participant families) (N = 52). Whole sample selection process is illustrated in Figure 1.

### 2.2. Inclusion Criteria

All patients who met the following criteria were included in this study; (1) aged between 15 and 60 years; (2) residing in the study area (Cantabria, Spain); (3) diagnosed with an FEP; (4) not having received antipsychotic treatment, and if so, not for more than six weeks; (5) meeting diagnostic criteria for brief psychotic disorder, schizophreniform disorder, schizophrenia, schizoaffective disorder, unspecified psychosis, or delusional disorder; (6) to have contactable first-degree relatives who openly agreed or rejected participation in the PAFIP-FAMILIAS study.

### 2.3. Ethics

Both the PAFIP and PAFIP-FAMILIAS projects were approved by the local institutional review committee (CEIm Cantabria) in accordance with international research ethics standards (approval numbers NCT0235832 and 2017.247). All participants were informed about the objectives of the study and gave their written consent. It is important to emphasize that no sociodemographic, neurocognitive, clinical, or genetic data from the relatives are handled in this work and only the decision of whether or not to participate in the PAFIP-FAMILIAS study was considered as a grouping criterion for the FEP patients. Therefore, the data on the patients’ neurocognitive outcomes, which are the focus of this study, come from the PAFIP study and are, thus, in compliance with the original informed consent of the PAFIP project. The PAFIP-FAMILIAS project allocated an economic compensation of EUR 50 to the relatives for covering expenses derived from the trip and the time in our neuropsychology lab. None of both studies received funding from any pharmaceutical company.

### 2.4. Sociodemographic Assessment

Professional neuropsychologists affiliated with the Mental Illness Research Department at Valdecilla Biomedical Research Institute (IDIVAL) in Santander, Spain conducted the interviews in the neuropsychology laboratory to collect essential socio-demographic and premorbid data. The data included age, education attainment, marital status, employment status, family history of psychosis, family socioeconomic status, residence area, and parental cohabitation status.

### 2.5. Clinical, Premorbid, and Neurocognitive Assessment

The assessments, performed by the same two professionals, lasted for approximately 2 h. The tests were organized into cognitive domains known to be consistently impacted in individuals with schizophrenia [28], including processing speed (assessed with the standard total score of the WAIS-III digit symbol subtest) [32], motor dexterity (assessed with the time to complete the Grooved Pegboard Test with the dominant hand) [33], working memory (assessed using the standard total score of the WAIS-III digits forward and backward subtests) [32], verbal memory (assessed with the Rey Auditory Verbal Learning Test) [34], visual memory (assessed by means of delayed recall of the Rey Complex Figure) [35], attention (assessed with the correct responses of the Continuous Performance Test) [36], and executive function (assessed with the trail B-A score of the Trail Making Test) [37]. Furthermore, consistent with previous research [38], a measure of global cognitive functioning (GCF) was calculated based on the obtained results, which reflected the extent of cognitive impairment, with higher scores indicating poorer functioning.

To estimate premorbid IQ (epIQ), defined as general cognitive capacity prior to illness onset, the WAIS-III Vocabulary subtest was utilized [32]. This measure has been shown to be a robust indicator of crystallized intelligence and is, therefore, less susceptible to the effects of illness interference [24]. Premorbid adjustment, defined as functioning and adaptation prior to illness onset considering five domains (social isolation, peer relations, academic performance, adaptation to school, and interests), was calculated with the global score of the Premorbid Adjustment Scale [39]. Chlorpromazine equivalent dosages (CED) were considered as previous works of the group [26]. Negative symptoms were assessed by means of the Scale for the Assessment of Negative Symptoms of Schizophrenia [40] and positive symptoms were assessed through the Scale for the Assessment of Positive Symptoms of Schizophrenia [41].

### 2.6. Data Analysis

Statistical analysis of the data was carried out using the Statistical Package for Social Science version 21.0 (SPSS Inc., Chicago, IL, USA). As a standard procedure, normality and homogeneity assumptions of variance (respectively, Kolmogorov–Smirnov and Levene tests) were conducted on each variable in order to select parametric or non-parametric tests for further analysis. Prior to the analysis steps, the sample was sorted based on eFI and sex, as previously described in the sample selection process. Subsequently, the influence of eFI and sex was assessed on each dependent variable through chi-square analysis for categorical variables and ANCOVA analysis for continuum variables. Whenever variables did not meet the assumptions, an ANCOVA non-parametric equivalent, Quade [42], was used (Appendix A). The following covariates were taken into account for each analysis: age, years of education, and epIQ. Bonferroni’s post hoc analysis was included to determine between which groups significant differences existed.

## 3. Results

### 3.1. Sample Description

The final sample consisted of 308 participants, of which 43.5% (N = 134) of their relatives agreed to participate in the PAFIP-FAMILIAS study, while 56.5% (N = 174) declined participation.

Regarding the sex of the patients, out of the 140 women, 62.9% (N = 88) of their relatives declined to participate, while 37.1% (N = 52) agreed. These percentages varied for men, with 51.2% (N = 86) of their relatives declining participation and 48.8% (N = 82) agreeing to participate.

### 3.2. Sociodemographic Findings

Regarding age, women with non-participant families had a significantly higher age (mean = 35 years) compared to the rest of the groups: men with non-participant families, men with participant families, and women with non-participant families (*p* ≤ 0.001). The age difference among men was also significant (*p* ≤ 0.05), with a higher age for men with non-participant families (mean = 29.53 years).

Speaking about years of education, the women’s groups exhibited statistically significant differences favoring them over the men’s groups. Both women with participant and non-participant families attained higher levels of education than their male counterparts (*p* ≤ 0.05). In addition, the findings indicated that women with non-participant families exhibited higher levels of educational attainment when compared to men with participant families. Similarly, women with participant families also displayed higher levels in comparison to men with non-participant families (*p* ≤ 0.05).

In terms of marital status, women’s groups presented significant differences when compared to men, especially with the group of men with participant families. Both participant and non-participant women were married to a greater extent than men with participant families (*p* ≤ 0.001). Similarly, men with non-participants presented a higher percentage of married individuals than men with participant families (*p* ≤ 0.005), yet it was still lower than the proportion observed for women with participant families (*p* ≤ 0.05). A trend was observed indicating that women with non-participant families were more likely to be married compared to their counterparts with participant families (*p* = 0.069).

In line with this, living arrangements also yielded interesting results. The most significant differences were found between women with non-participant families and men with participant families, with the latter group more likely to reside in their parents’ homes (*p* ≤ 0.001). Additionally, significant differences were observed in men with non-participant families, with a higher likelihood of living in their parents’ home than women with non-participant families (*p* ≤ 0.001), but still lower than that for men with participant families (*p* ≤ 0.05). Likewise, men with non-participant families presented higher rates of parental cohabitation status than women with non-participant families (*p* ≤ 0.05).

As for the family history of psychosis, unemployment, socioeconomic status, and residential area, no significant differences were found among the study groups.

All sociodemographic findings can be found in Table 1.

### 3.3. Clinical Premorbid, and Neurocognitive Findings

Verbal memory showed significant differences in favor of women with participant families compared to the other groups: men with non-participant families, women with non-participant families, and men with participant families (*p* ≤ 0.001; *p* ≤ 0.005; *p* ≤ 0.05, respectively).

According to processing speed, significant differences were found in favor of women with participant families compared to both men with participant and non-participant families (*p* ≤ 0.05).

Visual memory, on the other hand, showed results in favor of men compared to women with non-participant families: men with participant families and men with non-participant families (*p* ≤ 0.001; *p* ≤ 0.05, respectively).

The same applies to attention, where both groups of men performed better than women with non-participant families (*p* ≤ 0.05).

Although no significant differences were found in the other cognitive domains, favorable trends towards the group of women with participant families were observed regarding working memory, motor dexterity, and global deficit score when compared to non-participant families’ counterparts. At last, it is noteworthy that executive functions were the sole domain in which no significant differences were detected between any of the four groups. However, the same dominant trend of the study was observed in which the groups with participant families tended to perform better than those with non-participant families, albeit, as mentioned, without reaching statistical significance. There were no significant differences observed in relation to negative and positive symptomatology as well as premorbid adjustment.

All premorbid, clinical, and neurocognitive findings can be found in Table 2. Neurocognitive findings are illustrated in Figure 2.

## 4. Discussion

This research aimed to explore the relationship between eFI and the cognitive capacity in patients with FEP, using an indirect measure. The study provides insight into psychosocial factors that may be related to the cognitive functioning of these individuals and therefore their prognosis, according to Sheffield et al. (2018) [43].

The study specifically examines the impact of family support considering the family involvement in understanding and managing the disease, and its relationship with cognitive performance after a FEP, taking sex differences into consideration.

Results show that only 37.1% of female patients received positive responses from their families to participate in the PAFIP-FAMILIAS study, whereas the figure increased to 48.8% for male patients and younger age groups. The female group exhibited superior cognitive functioning, showcasing notable benefits potentially associated with FI, equaling men in visual memory and attention tasks, and outperforming other groups in verbal memory and processing speed.

The high rate of family rejection to participate observed in this study is consistent with prior research, where over a third of psychotic patients lacked family support despite having living relatives in 89.4% of cases [44].

These results may be due to the burden of caring for these patients, as reported by Chen et al. (2016) [23]. A lack of knowledge about the disease, objective or perceived isolation during treatment, and increased tasks and economic expenses can result in physical and mental burdens for relatives, who may eventually disconnect. Leal-Soto et al. (2012) showed that a higher objective burden resulted in less support from affected relatives [18].

Various explanations have been proposed for sex differences, which are currently under debate. Leung and Chue (2000) [45] suggest that men may receive less family support due to a high degree of conflict and fear among their families because of their higher aggression rates. Seeman (2012) [46], on the other hand, denies such differences in aggression. An alternative theory that could explain the data found in this article is related to the age of symptoms onset. The earlier onset of the disease in men [10,47], up to six years according to Seeman (2012) [46], would make it more likely for them to reside in their parents’ home during their debut and therefore have more family support. In contrast, women, with an increase in the age of symptom onset, would have a higher probability of having moved out before their debut, which could explain less daily contact with their family network.

This study provides evidence supporting the claim that family support is positively associated with better cognitive capacity in patients with FEP, particularly in women and specifically in verbal memory. The data reveal that the groups receiving family involvement were significantly younger in both men (M = 25 years vs. M = 29.53) and women (M = 29.22 years vs. M = 35).

Family relationships are some of the closest and more complex relationships people have [19]; therefore, the significance of these in molding an individual’s welfare and advancement cannot be overstated [48]. Furthermore, interpersonal stressors involving family members can elicit more profound emotional responses compared to those that do not involve kin [49]. Notably, the impact of family support on women could be attributed to differences in socialization between genders and the biological notion that women are more sensitive to social stimuli. The variance in socialization and communication patterns may clarify why women are more affected by the absence of family connections. Women respond to social stimuli to a greater extent than men, even from the early stages of their development [50]. These findings are relevant since women could base their cognitive development on more social aspects and therefore, feel more affected by the absence of these family relationships.

Likewise, gender roles could also contribute to the observed difference in the impact of family support. Research suggests that women are more likely to be involved in family dynamics [49] and that communication is a crucial element for socializing in women. These factors further underscore the importance of family support for women’s cognitive functioning.

Consequently, it should not come as a surprise that the domain showing the greatest difference in this study is verbal memory, as prior research suggests it is one of the cognitive variables most influenced by social support and relational contact [51]. Therefore, promoting a supportive network within the family unit that fosters social interaction and cognitive stimulation could result in improved cognitive functioning across multiple domains.

Furthermore, noteworthy observations of the absence of differences, such as those pertaining to residential areas or executive functions, warrant additional discussion within the context of this study.

It is important to note that Cantabria, the region in which the study was conducted, predominantly consists of rural communities, with only a few mid-size urban areas and three major urban centers, namely Santander, Torrelavega, and Laredo, housing the hospitals associated with the program. Additionally, during the study period from 2001 to 2018, a substantial portion of the population in Cantabria continued to reside in rural areas. Therefore, given the region’s predominantly rural nature and the inclusion of patients from the three major urban areas, it is unsurprising to observe a homogeneous sample in terms of residential areas. This inclusion likely contributed to sample balance and potentially mitigated the anticipated effects of the region’s rural composition.

The lack of differences among the study groups in terms of executive functions is a notable observation and thought-provoking observation that calls for further inquiry. This finding is particularly intriguing since the PAFIP-FAMILIAS study has addressed this topic finding familial differences in this domain [24].

### 4.1. Strengths and Limitations

It is noteworthy that the originality of this study lies in the exploration of the family involvement variable as a predictor of significant differences in the cognitive functioning of patients after their first psychotic episode, providing promising results in understanding the individual characteristics of these conditions. The election of an indirect measurement of this variable might avoid subjective biases inherent in self-report, and the inclusion of the sex variable provides interesting differences for understanding this phenomenon. However, at the same time, the use of a proxy approach to study FI may represent a limitation.

It is important to acknowledge that this study is not without limitations, among which are the following: its cross-sectional nature which we believe contributes to the understanding of the complex and bidirectional relationship between FI and mental health in patients with FEP but does not provide a causal explanation for the findings; the evolution or maintenance of these differences over time is unknown, as well as the tracking of the social support variable years before the psychotic episode; the duration of time elapsed between patients’ participation in the PAFIP program and the invitation of their family members to participate in the PAFIP-FAMILIAS study is not homogeneous. This is due to the fact that the PAFIP project started in 2001 and lasted for 17 years (until 2018), while the invitations to participate in PAFIP-FAMILIAS were initiated in the same year (2018). As a consequence, some participants may have received treatment through the PAFIP program up to 17 years before the recruitment for the PAFIP-FAMILIAS study began in 2018, while others may have been involved for only a few months since the program lasted for three years (with the last enrollees in 2018); the sample is limited to residents in Cantabria, which could affect the generalizability of the data by not having greater geographic or ethnic representativeness; lastly, the use of indirect measures to assess FI in order to avoid subjective biases may lead to other biases (but still offers an interesting point of view). It would be advisable to combine studies that openly inquire about both objective and perceived FI to achieve a greater understanding of the problem, since, despite being defended here as related, a single decision such as declining or agreeing to participate, is obviously unable to fully capture the whole nuances of FI in caregiving.

### 4.2. Further Lines of Research

Considering the knowledge provided by this study, several questions arise we consider deemed worthwhile endeavors to study in upcoming works.

Replication in diverse populations: The present study, conducted within the region of Cantabria, primarily comprised of rural communities (albeit with a balanced sample), offers a valuable foundation. However, future research endeavors would greatly benefit from extending the investigation to encompass other regions with distinct demographic compositions. Replicating the study in urban or metropolitan areas would provide valuable insights into potential variations in the observed patterns, thereby facilitating the establishment of generalizability. Similarly, expanding the study to fully rural communities would offer unique opportunities to enhance certain aspects, such as early diagnosis, which is likely to be more challenging in this population. Such efforts hold the potential to significantly improve treatment outcomes and contribute to the advancement of knowledge in the field.

Longitudinal studies: Adopting a longitudinal perspective would provide a deeper understanding of the temporal dynamics and developmental trajectories of in individuals with non-affective psychosis. This approach would allow for a nuanced examination of key factors, including, FI in treatment, caregiver burden, and neurocognitive impairment, among others. Through the systematic observation of these factors over time, researchers can assess how these abilities evolve and potentially identify critical periods for intervention and support.

Integration of objective and perceived FI: To foster a more comprehensive perspective on the caregiving experience, it would be advantageous to integrate studies that encompass both objective and perceived measures of FI. While this study emphasizes the significance of specific decisions, such as the choice to participate or decline, in capturing facets of FI, it is crucial to acknowledge that these singular determinations may not fully encapsulate the intricate complexity and subtleties of this multifaceted phenomenon. By incorporating assessments that encompass objective indicators of involvement (e.g., extent of engagement, frequency of interactions) and subjective perceptions of involvement (e.g., perceived support, burden perception), researchers can attain a more holistic comprehension of the dynamic interplay between individuals with non-affective psychosis and their families. Such an approach would facilitate an enriched understanding of the multifaceted nature of caregiving and contribute to the development of comprehensive frameworks for intervention and support.

Intervention and treatment strategies: Future research endeavors could center on the development and evaluation of targeted interventions aimed at enhancing and sustaining FI in the care of individuals with mental illness. By focusing on interventions that promote the continuity of FI and provide support, researchers can extend the benefits of such involvement in mitigating deficits and impairments. This line of inquiry holds promise for improving the overall well-being and functional outcomes of both individuals with mental illness and their families, contributing to a more comprehensive and effective approach to treatment and support.

## 5. Conclusions

FI has been found to be associated with better baseline cognitive functioning in patients with FEP, particularly for women. Enhanced cognitive function at illness onset is a critical factor for subsequent prognosis, and therefore, understanding the psychosocial factors that mediate these outcomes may lead to improved prevention in this area.

The caregiver’s burden that accompanies this illness often results in reduced or eliminated support, which can prove to be a significant disadvantage for affected patients, as here suggested.

New treatment approaches that promote FI, along with targeted interventions, would enhance the continuity of social support, reduce perceived burden, and promote the beneficial effects of such support for these patients.

## Figures and Tables

**Figure 1 healthcare-11-01902-f001:**
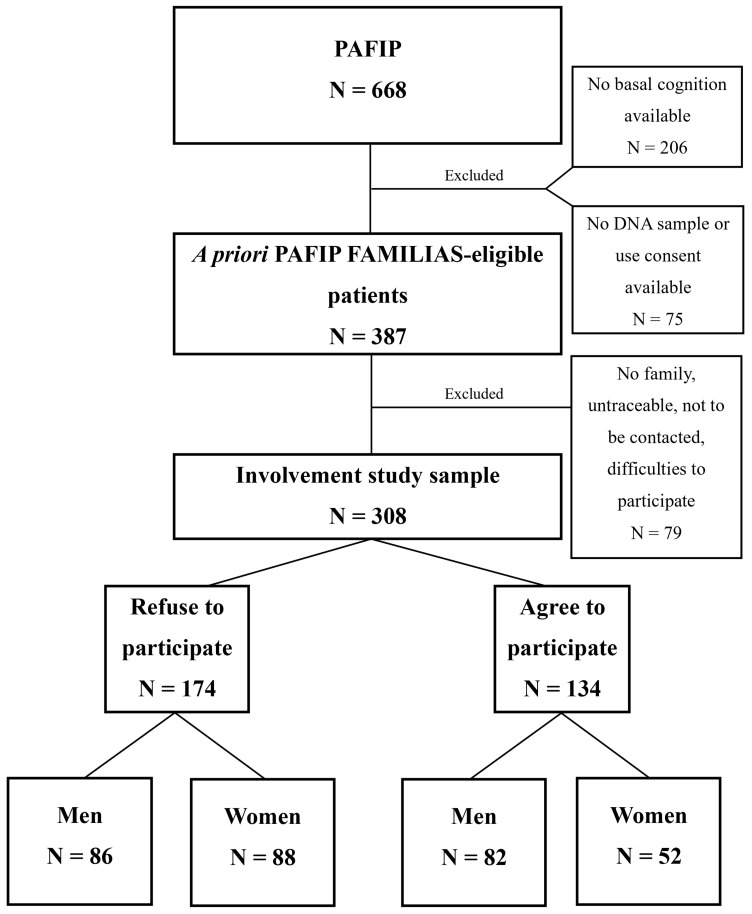
Family involvement study sample selection process.

**Figure 2 healthcare-11-01902-f002:**
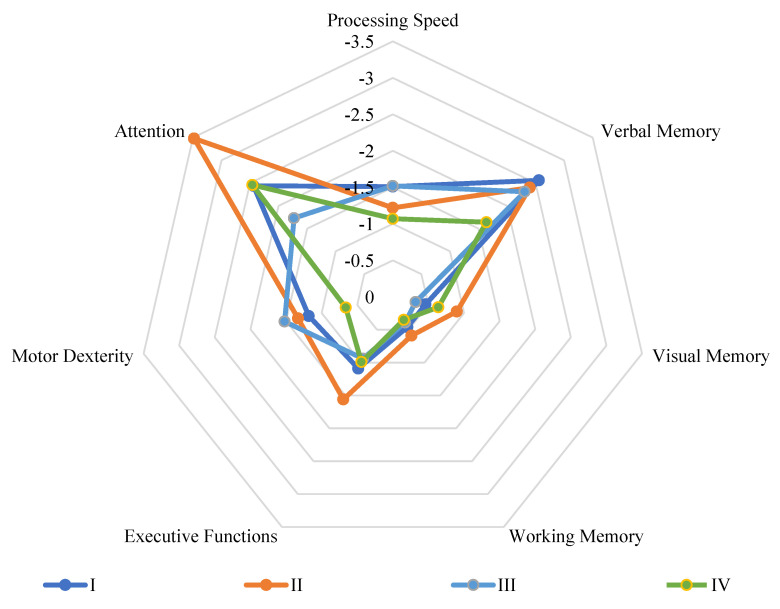
Comparisons of neurocognitive deficit profiles were conducted among the four study groups; men with non-participant families (I); women with non-participant families (II); men with participant families (III); women with participant families (IV). The statistical normality assumption (Z = 0) serves as the central point, with larger areas indicating greater neurocognitive deficits. Relevant fully statistically significant differences: those discerned between the same-sex homologous groups (i.e., participant, and non-participant), are delineated below. Specifically, verbal memory was found to be significantly greater in women with participant families than women with non-participant families (*p* ≤ 0.005), while relevant trends are also noted, including those pertaining to working memory (women with participant families > women with non-participant families, *p* = 0.088) and motor dexterity (*p* = 0.079). All means of measurement can be found in Section 2.

**Table 1 healthcare-11-01902-t001:** Sociodemographic comparisons between men and women whose families participated or refused participation in PAFIP-FAMILIAS study. All results covariate with age, years of education, and epIQ.

		PAFIP-FAMILIAS Non-Participant Families				PAFIP-FAMILIAS Participant Families					
		Men (I)		Women (II)		Men (III)		Women (IV)	Statistics	*p*-Value	Paired Comparisons
		(N = 86)		(N = 88)		(N = 82)		(N = 52)			
**Sociodemographics**	**N**	**Mean (SD)**	**N**	**Mean (SD)**	**N**	**Mean (SD)**	**N**	**Mean (SD)**	**F**		
Age	86	29.53 (8.13)	88	35.00 (9.93)	82	25.56 (7.05)	52	29.22 (9.36)	17.017	≤0.001	II > 1 ***; II > III ***; II > IV ***; I > III *
Years of education	86	9.85 (3.19)	86	11.40 (3.36)	81	9.98 (3.04)	52	11.69 (3.38)	6.020	≤0.001	IV > I *; II > I *; IV > III *; II > III *
	**N**	**N (%)**	**N**	**N (%)**	**N**	**N (%)**	**N**	**N (%)**	** *X* ** ** ^2^ **		
Marital status (married)	86	20 (23.25)	85	36 (42.35)	81	5 (6.17)	52	14 (26.92)	29.448	≤0.001	II > III ***; IV > III ***; I > III **; II > I *; II > IV (*p* = 0.069)
Family history of psychosis (yes)	86	22 (25.58)	88	20 (22.73)	81	16 (19.75)	52	11 (21.15)	0.876	0.831	
Unemployment (yes)	86	35 (40.70)	85	35 (41.18)	81	37 (45.68)	52	14 (26.92)	4.863	0.182	
Socioeconomic family status (low)	86	50 (58.14)	85	41 (48.24)	81	33 (40.74)	51	19 (37.25)	7.546	0.056	
Residence area (urban)	86	59 (68.81)	87	59 (67.82)	82	60 (73.17)	52	41 (78.85)	2.410	0.492	
Parental cohabitation status (yes)	86	47 (54.65)	85	33 (38.82)	81	57 (70.37)	52	28 (53.84)	16.641	≤0.001	III > II ***; III > I *; I > II *

epIQ: estimated premorbid intelligence quotient; * *p* ≤ 0.05; ** *p* ≤ 0.005; *** *p* ≤ 0.001. All means of measurement can be found in Section 2.

**Table 2 healthcare-11-01902-t002:** Clinical, premorbid, and neurocognitive comparisons between men and women whose families participated or refused participation in PAFIP-FAMILIAS study. All results covariate with age, years of education, and epIQ.

		PAFIP-FAMILIAS Non-Participant Families				PAFIP-FAMILIAS Participant Families					
		Men (I)		Women (II)		Men (III)		Women (IV)	Statistics	*p*-Value	Paired Comparisons
		(N = 86)		(N = 88)		(N = 82)		(N = 52)			
**Premorbid and clinical**	**N**	**Mean (SD)**	**N**	**Mean (SD)**	**N**	**Mean (SD**)	**N**	**Mean (SD)**	**F**		
epIQ	84	94.82 (14.78)	86	99.30 (12.44)	81	95.31 (14.13)	51	98.81 (13.47)	2.142	0.095	
PAS	82	2.83 (2.01)	82	2.90 (2.25)	78	3.35 (2.23)	49	3.33 (2.30)	0.592	0.620	
Negative symptoms	84	7.04 (6.84)	84	5.85 (5.28)	78	6.27 (6.34)	51	6.34 (6.13)	0.278	0.841	
Positive symptoms	84	12.93 (4.41)	84	14.43 (4.67)	79	13.94 (4.42)	51	15.38 (5.66)	2.202	0.088	
CED	84	188.97 (77.00)	84	204.04 (93.85)	80	194.11 (65.60)	51	207.81 (69.42)	0.542	0.654	
**Neuropsychological**	**N**	**Mean (SD)**	**N**	**Mean (SD)**	**N**	**Mean (SD)**	**N**	**Mean (SD)**	**F**		
Processing Speed	83	−1.51 (1.11)	84	−1.22 (0.95)	80	−1.52 (1.08)	51	−1.07 (1.03)	3.900	0.009	IV > III *; IV > I *
Verbal Memory	84	−2.56 (1.20)	84	−2.40 (1.33)	80	−2.31 (1.25)	51	−1.64 (1.34)	6.402	≤0.001	IV > I ***; IV > II **; IV > III *
Visual Memory	84	−0.46 (1.06)	84	−0.90 (0.92)	79	−0.32 (1.00)	51	−0.64 (0.92)	5.273	≤0.001	III > II ***; I > II *
Working Memory	84	−0.46 (0.73)	83	−0.59 (1.17)	80	−0.39 (0.80)	51	−0.35 (0.79)	2.901	0.035	IV > II (*p* = 0.088)
Executive Functions	83	−1.09 (2.18)	76	−1.56 (2.50)	78	−0.94 (1.66)	51	−0.99 (1.66)	1.298	0.276	
Motor Dexterity	83	−1.18 (1.59)	83	−1.33 (2.57)	80	−1.73 (3.68)	50	−0.66 (1.30)	2.432	0.065	IV > II (*p* = 0.079)
Attention	76	−2.44 (5.00)	84	−3.48 (4.58)	80	−1.73 (3.68)	47	−2.46 (3.30)	4.527	0.004	III > II *; I > II *
GDS	74	1.53 (1.07)	75	1.49 (0.85)	77	1.26 (0.93)	46	1.16 (0.85)	2.955	0.033	II > IV (*p* = 0.080)

epIQ: estimated premorbid intelligence quotient; PAS: premorbid adjustment scale; CED: chlorpromazine equivalent dosage; GDS: global deficit score; * *p* ≤ 0.05; ** *p* ≤ 0.005; *** *p* ≤ 0.001. All means of measurement can be found in Section 2.

## Data Availability

The data supporting the findings of this article are available upon request from the corresponding author, R.A.-A.

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
