# Peer review of "A Proxy Approach to Family Involvement and Neurocognitive Function in First Episode of Non-Affective Psychosis: Sex-Related Differences"

_healthcare, 2023, doi:10.3390/healthcare11131902_

Round 1

Reviewer 1 Report

Thank you for asking me to review this manuscript examining gender differences in the relationship between family support and neurocognitive function in individuals with first-episode psychosis (FEP). To that end, the authors conducted a cross-sectional post-hoc study based on the PAFIP cohort (a large epidemiological, 3-year longitudinal intervention program on FEP). The main findings were that higher family involvement (assessed by a proxy measure) was associated with better cognitive functioning (particularly verbal memory) and that this association was stronger in women than in men. The study would add to knowledge in the firld of caregiver burden in psychosis, but the manuscript has some major flaws that need to be addressed before it can be considered for publication.

MAJOR CONCERNS

A) Family support was assessed using a proxy measure (based on the relatives' decision to participate or not in the PAFIP-FAMILIAS study). The authors postulate that this indirect measure of family involvement is a strength of their study. In my opinion, it is an important limitation, as relatives could have refused to participate in the PAFIP-FAMILIAS study for reasons other than lack of support.

B) The authors state that family support has a significant predictive value for cognitive functioning in patients with FEP. However, as the study has a cross-sectional design, it is not possible to determine the direction of causality between family involvement and cognitive functioning. Therefore, I suggest that the authors report their findings in terms of association or relationship, but not causality.

C) The study does not examine the relationship between family support and other relevant variables such as positive symptoms (e.g. the role of paranoid delusions involving the family), negative symptoms, behavioural disturbances, number of psychotic relapses or substance use.

MINOR CONCERNS

a) Do not use subheadings in the introduction.

b) Authors should emphasise the importance of negative symptoms as central to the concept of schizophrenia.

c) The lack of differences between urban and rural areas deserves a comment.

d) A comment on why the groups did not differ in terms of executive functioning should also be included in the discussion.

The paper contains unusual expressions, such as “ Speaking about...” or “ isolation in treatment”.

Author Response

Dear Dr. José Antonio Monreal Ortiz

Thank you for considering our manuscript “Family involvement and neurocognitive function among first episode of non-affective psychosis: sex-related differences” (Healthcare-2374434). We would like to express our gratitude to the Reviewers for their constructive comments which have helped to clarify certain aspects of our work. Please find below a summary of our responses and note that any modifications made to the manuscript are highlighted in yellow.

Reviewer 1

The reviewer acknowledges the potential contribution of the study to the field of caregiver burden in psychosis, while suggesting that the authors further investigate and deliberate on the following points.

Comment #1. Family support was assessed using a proxy measure (based on the relatives' decision to participate or not in the PAFIP-FAMILIAS study). The authors postulate that this indirect measure of family involvement is a strength of their study. In my opinion, it is an important limitation, as relatives could have refused to participate in the PAFIP-FAMILIAS study for reasons other than lack of support.

Response: We greatly appreciate the constructive feedback and valuable contribution of the reviewer to our research. Indeed, we acknowledge that the proxy measure used to estimate family involvement may have limitations. We totally agree that there are many reasons why families may choose not to participate in a study, and we made sure to take this into account in our sample selection. Therefore, we considered exploring the usefulness of this proxy measure of family involvement by selecting those patients whose families declined or participated in the PAFIP-FAMILIAS study, excluding those patients whose families presented other reasons  for declining participation (work shifts, transportation and health-related issues). Due to the inability to establish the genuineness of such reported difficulties, we implemented this decision to minimize the risk of committing Type I error. The aim was to enhance the construct's validity and ensure that participation or non-participation more accurately reflected family involvement in the patients' mental health. Nonetheless, we understand that this may be a limitation, and therefore, we have revied and clarified the extent to which this proxy measure of family involvement may be valid.

Modifications have been made through several sections of the manuscript to address the noted limitations. These have been appropriately highlighted in yellow as follows:

Lines 109-119: Therefore, the primary objective is to explore the potential association between estimated levels of family involvement (FI) and neurocognitive function in patients with first episode of non-affective psychosis, while exploring any sex-based differences in this relationship. To minimize the reliance on self-report measures, a proxy measure approach will be utilized to estimate levels of received social support.

Refinement procedures have been implemented during the participant selection process to minimize potential biases and uncertainties inherent in the measure. The detailed description of these procedures is described in the methods section and illustrated in figure 1. Considering the scrupulous selection of participants and the subsequent measures undertaken to address the inherent limitations, it was deemed appropriate to proceed with the exploration of this measure within the scope of this study.

Lines 148-186: It is fundamental to acknowledge that proxy measures, by their nature, involve certain degree of imprecision, which can affect accuracy of the measure. To increase the internal validity of the measure, it is therefore advisable to reduce the risk of Type I error. In order to refine the conceptual classification criteria for the study's subjects, and to align them as closely as possible with the goal of acquiring comprehensive insight into their family member's illness while contributing to the existing knowledge in the field (thus, suggesting a stronger commitment to their family member's mental health), participants who, due to difficulties, were unable to accommodate convenient scheduling options for the PAFIP-FAMILIAS study (n=38), were excluded from the analyzed sample.

This decision was made due to our inability to verify the genuineness of the reported difficulties in participation, which primarily involved work shifts, transportation, or health-related issues. However, it is important to note that adequate provisions and facilities were provided to help address these concerns. The aim of this exclusion was to reduce the risk of falsely identifying relatives who were not involved in their loved one's care and to ensure to a higher degree that the included subjects truly represented the targeted population.

To support this decision, it is worth noting that study participants were afforded with flexible scheduling options to mitigate any possible inconvenience associated with study participation, which is particularly important in the context of caregiving. Furthermore, the study had a mean duration of no more than 2 hours and provided a compensation of 50€ for travel and parking expenses during the evaluation. Moreover, it is important to take into account that the study was conducted at a center equipped with ample parking facilities in close proximity and convenient access to abundant public transportation connections, including buses and trains.

Therefore, considering the aforementioned, despite there may be valid reasons for declining participation which do not necessarily reflect a lack of interest in their familiar's care, this procedure is believed to help mitigate this casuistry, remaining only those participants whose families agree to participate and those whose families decline participation due to more conviction-related reasons. Additionally, in the same step, subjects who could not be located despite repeated attempts to contact them (n=19), or who were conceptually or legally incompatible with the study's aim of examining the impact/correlation of family interest, were excluded. This comprises respectively patients who lacked family members (n=14) or those who requested not to be further contacted for more studies in order to respect their previous wishes (n=8).

In view of the facilities extended and the subsequent exclusion of subjects who, despite such measures, could not align their personal circumstances with study participation, the selection variable can be construed as reflecting a voluntary and uncoerced participation. This, in turn, enhances the representativeness of the sample in relation to the target population of individuals who are involved and non-involved in caregiving.

Lines 444-446: …to achieve a greater understanding of the problem, since, despite being defended here as related, a single decision such as declining or agreeing to participate, is obviously unable to fully capture the whole nuances of family involvement in caregiving.

Comment #2. The authors state that family support has a significant predictive value for cognitive functioning in patients with FEP. However, as the study has a cross-sectional design, it is not possible to determine the direction of causality between family involvement and cognitive functioning. Therefore, I suggest that the authors report their findings in terms of association or relationship, but not causality.

Response: We thank the reviewer for the comment. We fully acknowledge the importance of distinguishing between association and causality in the interpretation of our findings. We recognize that our cross-sectional study design does not allow us to establish a definitive causal direction between family involvement and cognitive functioning in patients with FEP. However, we believe that the findings of our study contribute to the understanding of the complex and bidirectional relationship between family involvement and mental health in patients with FEP. Following the reviewer's suggestion that our findings should be interpreted in terms of association and relationship rather than causality modifications in the terminology have been consistently applied.

The following modifications have been made in the manuscript and are highlighted in yellow:

Line 341-343: This research aims to explore the relationship of family involvement and the cognitive capacity in patients with first-episode psychosis, using an indirect measure. The study provides insight into psychosocial factors that may be related to the cognitive functioning…

Lines 351-352: The female group exhibited superior cognitive functioning, showcasing notable benefits associated with family involvement,

Lines 425-429: It is important to acknowledge that this study is not without limitations, among which is its cross-sectional nature which we believe contributes to the understanding of the complex and bidirectional relationship between family involvement and mental health in patients with FEP but does not provide causal explanation for the findings

Line 491: Family involvement has been found to be associated to better baseline cognitive…

Comment #3: The study does not examine the relationship between family support and other relevant variables such as positive symptoms (e.g. the role of paranoid delusions involving the family), negative symptoms, behavioral disturbances, number of psychotic relapses or substance use.

Response: We thank the reviewer for the comment. We appreciate the reviewer's suggestion to explore these relationships. We conducted analysis in several clinical variables. No significant differences where observed.

Results have been added and can be found highlighted in yellow in table 2 and in the manuscript as follows:

Lines 248-251: Negative symptoms were assessed by means of the Scale for the Assessment of Negative Symptoms of Schizophrenia [40] and positive symptoms were assessed through the Scale for the Assessment of Positive Symptoms of Schizophrenia [41].

Comment #4: Do not use subheadings in the introduction.

Response: We thank the reviewer for the comment. The manuscript will be changed in accordance.

Comment #5: Authors should emphasize the importance of negative symptoms as central to the concept of schizophrenia.

Response: We thank the reviewer for bringing this point to our attention. While our study primarily centered around investigating neurocognitive functioning, we highly value the reviewer's perspective and acknowledge the importance of negative symptomatology in schizophrenia research.

The manuscript has been modified as follows:

Lines 43-52: The symptomatic presentation of a psychotic episode is identified by a range of manifestations which encompasses the presence of positive and negative symptoms, either co-occurring or occurring independently [3,4]. The former entails the presence of delusions and hallucinations, along with aberrant motor behaviors and disorganized behavior. These symptoms contribute to the distortion of perception and reality experienced by individuals during a psychotic episode. The latter involve a reduction or loss in the ability to experience pleasure, emotion, motivation, and social communication. These symptoms can significantly impact an individual's daily functioning and overall well-being. While the ultimate causes are not yet fully understood, ongoing research within the scientific community continues to shed light on this complex matter.

Comment #6: The lack of differences between urban and rural areas deserves a comment.

Response: We thank the review for the comment. Indeed, the lack of differences regarding residence area deserves attention. In this regard, it is noteworthy to mention that Cantabria is predominantly a rural community with a few small urban areas and only three urban centers, such as Santander, Torrelavega, and Laredo, where hospitals are located (during the term of PAFIP, only Santander's hospital served as the hub of the program). Furthermore, much of the population in Cantabria continues to reside in rural areas or small urban centers, as was the case during the study period (2001-2018). Thus, given these factors, our sample can be viewed as an epidemiological sample of the region and therefore might explain the absence of statistical differences in the residential area. Considering this information, it would indeed be a captivating and worthwhile endeavor to investigate this matter further in upcoming studies and fully recognize the merit of the reviewer's suggestion.

The manuscript has been modified as follows:

Lines 402-413: Furthermore, noteworthy observations of absence of differences, such as those pertaining to residential area or executive functions, warrant additional discussion within the context of this study.

It is important to note that Cantabria, the region in which the study was conducted, predominantly consists of rural communities, with only a few mid-size urban areas and three major urban centers, namely Santander, Torrelavega, and Laredo, housing the hospitals associated with the program. Additionally, during the study period from 2001 to 2018, a substantial portion of the population in Cantabria continued to reside in rural areas. Therefore, given the region's predominantly rural nature and the inclusion of patients from the three major urban areas, it is unsurprising to observe a homogeneous sample in terms of residential area. This inclusion likely contributed to sample balance and potentially mitigated the anticipated effects of the region's rural composition.

Lines 450-460: Replication in diverse populations: The present study, conducted within the region of Cantabria, primarily comprised of rural communities (albeit with a balanced sample), offers a valuable foundation. However, future research endeavors would greatly benefit from extending the investigation to encompass other regions with distinct demographic compositions. Replicating the study in urban or metropolitan areas would provide valuable insights into potential variations in the observed patterns, thereby facilitating the establishment of generalizability. Similarly, expanding the study to fully rural communities would offer unique opportunities to enhance certain aspects, such as early diagnosis, which is likely to be more challenging in this population. Such efforts hold the potential to significantly improve treatment outcomes and contribute to the advancement of knowledge in the field.

Comment #7: A comment on why the groups did not differ in terms of executive functioning should also be included in the discussion.

Response: We thank the reviewer for the comment. We appreciate the suggestion to include information in the manuscript addressing the issue raised and we intend to follow through on it. Specifically, we will make sure to incorporate this point into our discussion section as suggested as well as highlighted mention in the results addressing the topic.

The manuscript has been modified as follows:

Lines 322-326: At last, it is noteworthy that executive functions were the sole domain in which no significant differences were detected between any of the four groups. However, the same dominant trend of the study was observed in which the supported groups tended to perform better than the unsupported ones, albeit, as mentioned, without reaching statistical significance.

Lines 414-417: The lack of differences among the study groups in terms of executive functions is a notable observation and thought-provoking observation that calls for further inquiring. This finding is particularly intriguing since the PAFIP-FAMILIAS study has addressed this topic finding familial differences in this domain [24].

Added references.

  1. Murillo-García, N., Díaz-Pons, A., Fernández-Cacho, L. M., Miguel-Corredera, M., Martínez-Barrio, S., Ortiz-García de la Foz, V., Neergaard, K., & Ayesa-Arriola, R. A family study on first episode of psychosis patients: Exploring neuropsychological performance as an endophenotype. Acta psychiatrica Scandinavica2022, 145(4), 384–396. DOI:10.1111/acps.13404
  2. Cannon-Spoor, H. E., Potkin, S. G., & Wyatt, R. J. Measurement of premorbid adjustment in chronic schizophrenia.Schizophrenia bulletin, 19828(3), 470–484. DOI:10.1093/schbul/8.3.470
  3. Andreasen N. C. Negative symptoms in schizophrenia. Definition and reliability. Archives of general psychiatry39(7), 784–788. 1982, DOI:10.1001/archpsyc.1982.04290070020005
  4. Andreasen, N. C., Arndt, S., Miller, D., Flaum, M., & Nopoulos, P. Correlational studies of the Scale for the Assessment of Negative Symptoms and the Scale for the Assessment of Positive Symptoms: an overview and update.Psychopathology, 199528(1), 7–17. DOI:10.1159/000284894

Reviewer 2 Report

The authors report the results of the study aimed to investigate sex differences in neurocognitive outcomes in patients with a first episode non-affective psychosis basing on family insolvent. The study may have presented interesting results and contribute to the understanding of this important topic. However, there are major conceptual and possible ethical concerns that should be addressed in the manuscript and as some clarifications associated with the experimental design and data presentation are needed, as well. Nevertheless, I believe that the issues could be addressed by changes in the manuscript including appropriate definitions and method descriptions. Thus, to further evaluate the study, the following issues should be addressed, as they may affect results interpretations of the results and conclusions:

11.    First of all, the major conceptual issue is that authors substitute the “family involvement (FI)” with the interest in participation in the PAFIP-FAMILIAS study. Notably, in the 1.3. Study contributions, the authors define FI as follows: “ … the present study seeks to explore, through a non-self-reported measure, a component of family support such as FI: defined as interest in the illness and involvement in understanding it. … “ (95-97), and further, in the same paragraph, the authors describe a non-self-reported measure of FI as:”… For this purpose, the decision to participate or not in the PAFIP-FAMILIAS study on relatives of patients with FEP is used as a non-self-reported measure. (97-99), which contradicts both the authors’ FI definition (95-96) and the definition of family support by Leal-Soto et al. (2012) (see Introduction, 68-78). Obviously, a refusal of participation in a particular study is not a measure of any aims or intension why that study was created and performed. Likely, there were family members that are “interested in the illness and involvement in understanding it” and provide family support to extend of their abilities, but refused to participate in the study because of other reasons including lack of trust to the study and/or provider. Thus, to resolve this major conceptual flaw and possible ethical issues, the “family involvement (FI)” (as well as related terms “supported” and “unsupported”) needs to be substituted with the status of decision or participation/non-participation in the PAFIP-FAMILIAS throughout the manuscript including title and abstract.

22.   In the Discussion, 4.1. Strengths and limitations, the authors wrote that they attempted to explore “the family involvement variable as a predictor of significant differences in the cognitive functioning of patients after their first psychotic episode, providing promising results in understanding the individual characteristics of these conditions”(312-315) and, in the Conclusions, that “Family involvement has been found to be related to better baseline cognitive functioning in patients with first-episode psychosis” (328-329) indirectly suggesting a causal relationship of the family involvement, which is actually the decision to participate in other study (see the comment above), and cognitive deficits. However, the study design allows to suggest only associative relationship between these parameters. Thus, revisions and clarifications should be included in the Discussion as well as in the Conclusions.

33.    Regarding a possible ethical concern, the informed consent(s) of the participant should be explicitly clarified in the Method section as well as un the Informed Consent Statement. Presumably, the subjects’ neurocognitive outcome data are a part of the PAFIP study. However, the 2.3. Ethics mentioned both PAFIP and PAFIP-FAMILIAS studies and that “All participants … gave their written consent” (149-152). According to the study design, the subject data were reported according with the participation in the PAFIP-FAMILIAS study, in which majority of the eligible subjects’ relatives refused to participate. Thus, an explicit statement from the authors that those subject (who refused participate) provided a consent to use the data or such data reporting is in compliance with the original informed consent of the PAFIP study.

44.    As the PAFIP study lasted about 17 years, the quantitative information about time of between subjects’ involvement in the PAFIP study and invitation for PAFIP-FAMILIAS study should be included and discussed in the manuscript.

55.    Reasons of rejection to participate in the particular PAFIP-FAMILIAS study should be included in the discussion in addition to discussion of published data on the general reasons of “family isolation” (264-271), as these two concepts are not congruent (see comments above).

65.    The method section should include information more detailed information on “premorbid data” collection and “premorbid data” definition.

76.    Although an approach for the statistical analyses looks overall appropriate, some additional clarifications are needed including justification of normality assumption for the particular outcomes reported in the manuscript and specific indication of specific statistical test used for each reported p values in the text of the manuscript including figure/table legends.

87.    The Figure 2 and/or its legend should include detailed description of the data presentation and show statistically significant differences either in text form or graphically (in the figure).

Overall quality of English is appropriate. However, some revisions of writing needed for more clear description of the concepts. Another minor issue: Figure 1 contains a Spanish word.

Author Response

Dear Dr. José Antonio Monreal Ortiz

Thank you for considering our manuscript “Family involvement and neurocognitive function among first episode of non-affective psychosis: sex-related differences” (Healthcare-2374434). We would like to express our gratitude to the Reviewers for their constructive comments which have helped to clarify certain aspects of our work. Please find below a summary of our responses and note that any modifications made to the manuscript are highlighted in yellow.

Reviewer 2

The reviewer has acknowledged the possible contribution of the study in the investigation of gender differences in neurocognitive outcomes in patients with non-affective first episode psychosis based on family involvement. The reviewer recognizes that the study has yielded interesting results, which may aid in the comprehension of this crucial topic. However, the reviewer has also wisely observed some noteworthy issues related to concept, possible ethical concerns, and the presentation of findings that warrant careful consideration. Nonetheless, the reviewer maintains that these concerns can be addressed effectively with appropriate modifications to the manuscript, including the provision of precise definitions and descriptions of methods. In light of these issues, the reviewer recommends that the authors address the concerns in the manuscript to ensure that the interpretation of results and conclusions is accurate.

Comment #1. First of all, the major conceptual issue is that authors substitute the “family involvement (FI)” with the interest in participation in the PAFIP-FAMILIAS study. Notably, in the 1.3. Study contributions,the authors define FI as follows: “ … the present study seeks to explore, through a non-self-reported measure, a component of family support such as FI: defined as interest in the illness and involvement in understanding it. … “ (95-97), and further, in the same paragraph, the authors describe a non-self-reported measure of FI as:”… For this purpose, the decision to participate or not in the PAFIP-FAMILIAS study on relatives of patients with FEP is used as a non-self-reported measure. (97-99), which contradicts both the authors’ FI definition (95-96) and the definition of family support by Leal-Soto et al. (2012) (see Introduction, 68-78). Obviously, a refusal of participation in a particular study is not a measure of any aims or intension why that study was created and performed. Likely, there were family members that are “interested in the illness and involvement in understanding it” and provide family support to extend of their abilities, but refused to participate in the study because of other reasons including lack of trust to the study and/or provider. Thus, to resolve this major conceptual flaw and possible ethical issues, the “family involvement (FI)” (as well as related terms “supported” and “unsupported”) needs to be substituted with the status of decision or participation/non-participation in the PAFIP-FAMILIAS throughout the manuscript including title and abstract.

Response: We hold in high regard the reviewer for this attention to detail and for the opportunity to explain this conceptual decision.We acknowledge that our definition of family involvement (FI) may have caused confusion, as our proposed approach for FI relied on the participation of families in the PAFIP-FAMILIAS study as a proxy measure of this construct. While we believe that this approach has merits, we agree that this measure may not be sufficiently comprehensive to fully capture the nuanced intricacies of family involvement.

Our exploratory approach suggestion is that families who participate in a study focused on mental health may have a higher predisposition to be involved in the care of their ill familiars. Conversely, a negative response to study participation may indicate a lesser attachment to caregiving. We would like to emphasize that flexible scheduling options were offered to participants to minimize potential inconveniences associated with study participation and scheduling conflicts, which is particularly important in the context of caregiving. We understand that there could be many legitimate reasons why families may choose not to participate in a study, and we have taken this into account in our sample selection.

Thus, we selected patients whose families participated in the PAFIP-FAMILIAS study, excluding those whose families presented other reasons for declining participation but we were unable to verify (primarily involved work shifts, transportation, or health-related issues). While this procedure may exclude participants, including some those whose families are actually involved in understanding their loved one's illness but have decided not to participate due to the alleged circumstances, we believe it offers a higher level of construct  internal validity in accurately reflecting family involvement in patients' mental health minimizing the risk of error Type I commission.

However, we understand that this may be a limitation, and thus in the discussion and conclusions, we will review and clarify the extent to which this proxy measure of family involvement may be valid. We also appreciate the reviewer's suggestion to use the terms "participation" and "non-participation" to describe the family's decision and we will consider changing the terminology used to describe the groups.

Finally, we hope the explanation provided clarifies the rationale behind our choice of the FI terminology for the title. We will make the necessary adjustments in the manuscript and in the tittle to ensure that our definition of FI is clear and precise. Once again, we express our gratitude to the reviewer for their invaluable comment, and we sincerely hope that we have addressed their concerns appropriately.

The following modifications have been made to take all these suggestions into account and are presented in yellow highlight:

Lines 2-4: A proxy approach to family involvement: sex differences and neurocognitive function among first episode of non-affective psychosis

Lines 148-186: It is fundamental to acknowledge that proxy measures, by their nature, involve certain degree of imprecision, which can affect accuracy of the measure. To increase the internal validity of the measure, it is therefore advisable to reduce the risk of Type I error. In order to refine the conceptual classification criteria for the study's subjects, and to align them as closely as possible with the goal of acquiring comprehensive insight into their family member's illness while contributing to the existing knowledge in the field (thus, suggesting a stronger commitment to their family member's mental health), participants who, due to difficulties, were unable to accommodate convenient scheduling options for the PAFIP-FAMILIAS study (n=38), were excluded from the analyzed sample.

This decision was made due to our inability to verify the genuineness of the reported difficulties in participation, which primarily involved work shifts, transportation, or health-related issues. However, it is important to note that adequate provisions and facilities were provided to help address these concerns. The aim of this exclusion was to reduce the risk of falsely identifying relatives who were not involved in their loved one's care and to ensure to a higher degree that the included subjects truly represented the targeted population.

To support this decision, it is worth noting that study participants were afforded with flexible scheduling options to mitigate any possible inconvenience associated with study participation, which is particularly important in the context of caregiving. Furthermore, the study had a mean duration of no more than 2 hours and provided a compensation of 50€ for travel and parking expenses during the evaluation. Moreover, it is important to take into account that the study was conducted at a center equipped with ample parking facilities in close proximity and convenient access to abundant public transportation connections, including buses and trains.

Therefore, considering the aforementioned, despite there may be valid reasons for declining participation which do not necessarily reflect a lack of interest in their familiar's care, this procedure is believed to help mitigate this casuistry, remaining only those participants whose families agree to participate and those whose families decline participation due to more conviction-related reasons. Additionally, in the same step, subjects who could not be located despite repeated attempts to contact them (n=19), or who were conceptually or legally incompatible with the study's aim of examining the impact/correlation of family interest, were excluded. This comprises respectively patients who lacked family members (n=14) or those who requested not to be further contacted for more studies in order to respect their previous wishes (n=8).

In view of the facilities extended and the subsequent exclusion of subjects who, despite such measures, could not align their personal circumstances with study participation, the selection variable can be construed as reflecting a voluntary and uncoerced participation. This, in turn, enhances the representativeness of the sample in relation to the target population of individuals who are involved and non-involved in caregiving.

Lines 190-195: … study (hereafter referred to as men with non-participant families) (N= 86), FEP women whose families refused to participate in PAFIP FAMILIAS study (women with non-participant families) (N=88), FEP men whose families agreed to participate in PAFIP FAMILIAS study (men with participant families) (N= 82), and FEP women whose families agreed to participate PAFIP FAMILIAS study (women with participant families) (N=52).

Lines 442-446: It would be advisable to combine studies that openly inquire about both objective and perceived family involvement to achieve a greater understanding of the problem, since, despite being defended here as related, a single decision such as declining or agreeing to participate, is obviously unable to fully capture the whole nuances of family involvement in caregiving.

Lines 468-481: Integration of objective and perceived family involvement: To foster a more comprehensive perspective on the caregiving experience, it would be advantageous to integrate studies that encompass both objective and perceived measures of family involvement. While this study emphasizes the significance of specific decisions, such as the choice to participate or decline, in capturing facets of family involvement, it is crucial to acknowledge that these singular determinations may not fully encapsulate the intricate complexity and subtleties of this multifaceted phenomenon. By incorporating assessments that encompass objective indicators of involvement (e.g., extent of engagement, frequency of interactions) and subjective perceptions of involvement (e.g., perceived support, burden perception), researchers can attain a more holistic comprehension of the dynamic interplay between individuals with non-affective psychosis and their families. Such an approach would facilitate an enriched understanding of the multifaceted nature of caregiving and contribute to the development of comprehensive frameworks for intervention and support.

Comment #2: In the Discussion, 4.1. Strengths and limitations, the authors wrote that they attempted to explore “the family involvement variable as a predictor of significant differences in the cognitive functioning of patients after their first psychotic episode, providing promising results in understanding the individual characteristics of these conditions”(312-315) and, in the Conclusions, that “Family involvement has been found to be related to better baseline cognitive functioning in patients with first-episode psychosis” (328-329) indirectly suggesting a causal relationship of the family involvement, which is actually the decision to participate in other study (see the comment above), and cognitive deficits. However, the study design allows to suggest only associative relationship between these parameters. Thus, revisions and clarifications should be included in the Discussion as well as in the Conclusions.

Response: We appreciate and accept the reviewer's insightful feedback regarding the need to clarify the relationship between family involvement and cognitive deficits in our study. We agree that our study design only permits us to suggest an association between these parameters rather than causality. Therefore, we will revise our discussion and conclusions to accurately reflect this limitation and provide a detailed interpretation of our findings in terms of relationship and association. We greatly appreciate the reviewer's constructive criticism, and we believe that these revisions will strengthen the quality and validity of our research.

The manuscript has been modified as follows:

Lines 341-343: This research aims to explore the relationship of family involvement and the cognitive capacity in patients with first-episode psychosis, using an indirect measure. The study provides insight into psychosocial factors that may be related to the cognitive functioning…

Lines 425-428: It is important to acknowledge that this study is not without limitations, among which are the following: its cross-sectional nature which we believe contributes to the understanding of the complex and bidirectional relationship between family involvement and mental health in patients with FEP but does not provide causal explanation for the findings

Comment #3: Regarding a possible ethical concern, the informed consent(s) of the participant should be explicitly clarified in the Method section as well as un the Informed Consent Statement. Presumably, the subjects’ neurocognitive outcome data are a part of the PAFIP study. However, the 2.3. Ethics mentioned both PAFIP and PAFIP-FAMILIAS studies and that “All participants … gave their written consent” (149-152). According to the study design, the subject data were reported according with the participation in the PAFIP-FAMILIAS study, in which majority of the eligible subjects’ relatives refused to participate. Thus, an explicit statement from the authors that those subject (who refused participate) provided a consent to use the data or such data reporting is in compliance with the original informed consent of the PAFIP study.

Response: We thank the reviewer bringing up this important point regarding the ethical concerns raised in our study. We would like to clarify that the data used in our study is in accordance with the original informed consent of the PAFIP study. Our study only uses the PAFIP-FAMILIAS study as a classification criterion for grouping the subjects, and the data on the subjects' neurocognitive outcomes are part of the PAFIP study. We will make sure to explicitly mention this in the Method section of our manuscript. We understand the importance of ethical considerations in research and will ensure in the manuscript that our study adheres to the highest ethical standards. Thank you for bringing this issue to our attention, and we hope that this explanation addresses your concern.

The manuscript has been modified as follows:

Lines 210-216: It is important to emphasize that no sociodemographic, neurocognitive, clinical, or genetic data from the relatives is handled in this work and only the decision of whether or not to participate in the PAFIP-FAMILIAS study was considered as a grouping criterion for the FEP patients. Therefore, the data on the patients' neurocognitive outcomes, which is the focus of this study, comes from the PAFIP study and is, thus, in compliance with the original informed consent of the PAFIP project.

Comment #4: As the PAFIP study lasted about 17 years, the quantitative information about time of between subjects’ involvement in the PAFIP study and invitation for PAFIP-FAMILIAS study should be included and discussed in the manuscript.

Response: We esteem the reviewer for their insightful comment on our work regarding the inclusion of quantitative information about the time between subjects' involvement in the PAFIP study and their invitation to participate in the PAFIP-FAMILIAS study. We acknowledge the importance of addressing this aspect, as it may impact the interpretation of our findings. As the reviewer correctly pointed out, some participants may have received treatment through the PAFIP program up to 17 years before the recruitment of the PAFIP-FAMILIAS study was initiated in 2018, while others may have only been involved for a few months, as the program lasted for three years (with the last enrollees in 2018). In response to your feedback, we will make sure to provide a detailed discussion of this information in the revised manuscript. We appreciate your contribution to the refinement of our work.

The manuscript has been modified as follows:

Lines 430-438: the duration of time elapsed between patients' participation in the PAFIP program and the invitation of their family members to participate in the PAFIP-FAMILIAS study is not homogeneous. This is due to the fact that the PAFIP project started in 2001 and lasted for 17 years (until 2018), while the invitations to participate in PAFIP-FAMILIAS were initiated in the same year(2018). As a consequence, some participants may have received treatment through the PAFIP program up to 17 years before the recruitment for the PAFIP-FAMILIAS study began in 2018, while others may have been involved for only a few months, since the program lasted for three years (with the last enrollees in 2018)

Comment #5: Reasons of rejection to participate in the particular PAFIP-FAMILIAS study should be included in the discussion in addition to discussion of published data on the general reasons of “family isolation” (264-271), as these two concepts are not congruent (see comments above).

Response: We thank the reviewer for bringing up this important point that allows us both to better clarify conceptual classification criteria as well as declining-participation reasons. We agree that it is crucial to provide a thorough discussion of the reasons for rejection to participate in the PAFIP-FAMILIAS study, in addition to the published data on the general reasons for "family isolation" that we previously discussed in our manuscript. We understand that these two concepts may not be congruent and that it is important to distinguish between them. In response to your feedback, we will include a comprehensive discussion of the reasons for rejection to participate in the PAFIP-FAMILIAS study. We will carefully review our data and make sure that our manuscript accurately reflect these differences. Once again, we appreciate your insightful comments and suggestions, and we believe that this revision will greatly improve the clarity and accuracy of our manuscript.

See response to comment #1.

Comment #6: The method section should include information more detailed information on “premorbid data” collection and “premorbid data” definition.

Response: We thank the reviewer for the comment on our method section. We appreciate his/her feedback and have taken it into consideration in order to improve the clarity and rigor of our study. In regards to the collection and definition of premorbid data, we agree that additional details should be included in the method section to provide a more comprehensive understanding of our approach. We will make sure to include a more detailed description of the process we used to collect and define premorbid data, including the specific instruments and measures employed. Moreover, we will clarify the definition of premorbid data, emphasizing its significance as a measure of an individual's cognitive functioning prior to the onset of psychosis, and how this information can help to better understand the relationship between early cognitive deficits and subsequent mental health outcomes. Once again, thank you for this valuable feedback, and we hope that the revised method section will meet your expectations and contribute to the scientific rigor of our study.

The manuscript has been modified as follows:

Lines 243-248: To estimate premorbid IQ, defined as general cognitive capacity prior to illness onset, the WAIS-III Vocabulary subtest was utilized [32]. This measure has been shown to be a robust indicator of crystalized intelligence and is therefore less susceptible to the effects of illness interference [24]. Premorbid adjustment, defined as functioning and adaptation in prior to illness onset considering five domains (social isolation, peer relations, academic performance, adaptation to school, and interests), was calculated with the global score of the Premorbid Adjustment Scale [39].

Comment #7: Although an approach for the statistical analyses looks overall appropriate, some additional clarifications are needed including justification of normality assumption for the particular outcomes reported in the manuscript and specific indication of specific statistical test used for each reported p values in the text of the manuscript including figure/table legends.

Response: We respectfully thank the reviewer for its valuable feedback and suggestions regarding this issue and we appreciate the interest and the time spent on reviewing our manuscript. Regarding the concerns on the justification of the normality assumption and the specific indication of the statistical tests used for each reported p value, we have taken your comments seriously and have revised the manuscript accordingly.

To address the normality assumptions issue, we would like to emphasize that, as stated in the text in the data analysis paragraph (lines196-199), we conducted normality and homogeneity assumptions of variance tests with Kolmogorov-Smirnov and Levene tests, respectively on each variable to select appropriate parametric or non-parametric tests for further analysis. While we recognize the importance of transparency in reporting statistical tests, we believe that including such information in the tables is sufficient for the readers to replicate our analyses.

We have introduced the following clarifications to the manuscript, which are now highlighted in yellow for easy identification:

Lines 254-256: As a standard procedure, normality and homogeneity assumptions of variance (respectively Kolmogorov-Smirnov and Levene tests)

To address the issue regarding the statistical tests used for each reported p value in the text of the manuscript including figure/table legends, while we acknowledge the importance of providing as much information as possible for the scientific community, we would like to kindly defend the initial decision to exclude detailed statistical information from the body of the text to ensure clarity and ease of reading. We understand the reviewer's concerns but believe that the inclusion of additional statistical information in the main text may disrupt the flow of the content of the manuscript.

Nonetheless, we fully comprehend and acknowledge the reviewer's standpoint that it is crucial to offer an adequate amount of information for the scientific community. However, given that all statistical tests conducted have been clearly presented in the tables in the statistics column, with all due respect, we kindly request adhering to the original presentation of the results in the main text.

Once again, we thank the reviewer's valuable comments and suggestions, and hope that our explanations meets their expectations.

Comment #8: The Figure 2 and/or its legend should include detailed description of the data presentation and show statistically significant differences either in text form or graphically (in the figure).

Response: We apreciate the reviewer for this valuable feedback regarding the data presentation on our manuscript. We appreciate your attention to detail and agree that it is important to provide a clear and comprehensive description of the data presented in Figure 2.

This figure displays comparisons of neurocognitive deficit profiles among the four study groups, with the statistical normality assumption (Z=0) as the central point. We acknowledge that the inclusion of detailed descriptions of the data presentation and statistically significant differences would greatly enhance the figure and its accompanying legend.

Therefore, we have revised the legend of Figure 2 to provide a more detailed description of the data presented and have included information on the statistically significant differences between the same-sex homologous groups (i.e., family-involved and family-non-involved) in the text bellow the figure. We hope that these changes address your concerns and improve the clarity of our manuscript. Once again, we thank you for your constructive feedback and suggestions.

The manuscript has been modified accordingly to the suggestions and is now presented as follows:

Lines 332-339: Figure 2. Comparisons of neurocognitive deficit profiles were conducted among the four study groups; men with non-participant families (I); women with non-participant families (II); men with participant families (III); women with participant families (IV). The statistical normality assumption (Z=0) serves as the central point, with larger areas indicating greater neurocognitive deficits. Relevant fully statistically significant differences: those discerned between the same-sex homologous groups (i.e., participant, and non-participant), are delineated below. Specifically, verbal memory was found to be significantly greater in women with participant families than women with non-participant families (p≤0.005), while relevant trends are also noted, including those pertaining to working memory (women with participant families > women with non-participant families, p=0.088) and motor dexterity (p=0.079).

Added references.

  1. Murillo-García, N., Díaz-Pons, A., Fernández-Cacho, L. M., Miguel-Corredera, M., Martínez-Barrio, S., Ortiz-García de la Foz, V., Neergaard, K., & Ayesa-Arriola, R. A family study on first episode of psychosis patients: Exploring neuropsychological performance as an endophenotype. Acta psychiatrica Scandinavica2022, 145(4), 384–396. DOI:10.1111/acps.13404
  2. Cannon-Spoor, H. E., Potkin, S. G., & Wyatt, R. J. Measurement of premorbid adjustment in chronic schizophrenia.Schizophrenia bulletin, 19828(3), 470–484. DOI:10.1093/schbul/8.3.470
  3. Andreasen N. C. Negative symptoms in schizophrenia. Definition and reliability. Archives of general psychiatry39(7), 784–788. 1982, DOI:10.1001/archpsyc.1982.04290070020005
  4. Andreasen, N. C., Arndt, S., Miller, D., Flaum, M., & Nopoulos, P. Correlational studies of the Scale for the Assessment of Negative Symptoms and the Scale for the Assessment of Positive Symptoms: an overview and update.Psychopathology, 199528(1), 7–17. DOI:10.1159/000284894

Round 2

Reviewer 2 Report

I would like to thank the authors for paying attention to my comments and steps performed to improve the manuscript. However, there are still two conceptual flaws that require further major revisions of the manuscript: (1) misconception of the term “family involvement” and (2) inadequate description/justification of the statistical analyses. Of note, the reason of request for the detailed description for statistics in the original comments, was because the results were unclear, and the analyses did not appear to be appropriate according to the description that may rise questions about validity of the results and their interpretation. More details are described below.

1.      First of all, the participation/non-participation in a particular study cannot be considered in any means as proxy/surrogate measure of the family involvement. At minimum, it should be clearly stated in the abstract and in the limitations sub-section of discussion.

2.      Terms “family involvement (FI)”, “family support (FS)” and “participant/non-participant families“ are used interchangeably through the manuscript (for example, in 2.6. Data analysis the authors wrote: “The influence of FS and sex was assessed …” (257), whereas there was no variable such as FS included in the results). Moreover, even the abbreviations were introduced for some of these terms, non-abbreviated forms “family involvement” and “family support” are used in addition to abbreviated ones, which may lead to impression that meaning of these terms are different in a particular context. Thus, revisions throughout the manuscript are further required.

3.      Although the authors argue that the data are described enough to allow to reproduce the analyses, providing an appropriate description of the analyses in the Methods section is critical to also access whether the reported results are correctly analyzed and lead to conclusions described in the manuscript. Currently, this information is missing. As an option, the details could be included in supplementary materials, is the authors concern that “inclusion of additional statistical information in the main text may disrupt the flow of the content of the manuscript.”

4.      The authors wrote that the “normality” assumption was tested as “a standard procedure” [“As a standard procedure, normality and homogeneity assumptions of variance (respectively Kolmogorov-Smirnov and Levene tests) were conducted on each variable in order to select parametric or non-para- metric tests for further analysis.”(254-257)]. However, the normality assumption and parametric tests are not optimal or even inappropriate (unless justified by the authors) for several data types such as PAS, GDS, SAPS, SANS, etc., even, IQ and CED. 

5.      P numbers reported in the tables and figures are confusing because they are not linked to descriptions of specific tests performed.

6.      Comparing non-statistically significant data does not provide any meaningful information (e.g., II > IV (p = 0.080, etc. in Table 2.)

The overall English is acceptable, but extensive revision of writing is required to clarify the terminology.

Author Response

Dear Dr. José Antonio Monreal Ortiz

Thank you for considering our manuscript A proxy approach to family involvement and neurocognitive function in first-episode of non-affective psychosis: sex-related differences previously titled “Family involvement and neurocognitive function among first episode of non-affective psychosis: sex-related differences” (Healthcare-2374434) for publication. We would like to express our gratitude to the Reviewer for their constructive comments which have helped to clarify certain aspects of our work. Below, we provide a comprehensive summary of our responses, emphasizing any revisions made to the manuscript by highlighting them in yellow.

The reviewer thanks the authors for their consideration of the previous suggestions aimed at enhancing the manuscript's quality. Nevertheless, the reviewer identifies two remaining conceptual flaws that necessitate further substantial revisions to the manuscript.

Comment #1: First of all, the participation/non-participation in a particular study cannot be considered in any means as proxy/surrogate measure of the family involvement. At minimum, it should be clearly stated in the abstract and in the limitations sub-section of discussion.

Response #1: We greatly appreciate the constructive feedback and valuable contribution of the reviewer to our research. Indeed, we acknowledge the potential limitations associated with employing a proxy measure to estimate family involvement, as we recognize that it cannot, by any means, replace the use of more direct measures typically encompassing self-reported approaches, for capturing this multidimensional aspect.

In light of these considerations, we propose this approach as it may serve as valuable tools for capturing specific dimensions of the variable that may be susceptible to the influence of social desirability bias. This becomes particularly salient considering the potential challenges individuals may encounter when openly acknowledging a decline in their involvement in caring for an ill loved one. Proxy measures have the potential to provide complementary information in the absence of direct measures.

These considerations have been previously addressed and discussed within several sections of last version of the manuscript (methods, limitations, and  further lines of research) as follows:

Lines 150-151: It is fundamental to acknowledge that proxy measures, by their nature, involve certain degree of imprecision, which can affect accuracy of the measure.

Lines 445-447: …despite being defended here as related, a single decision such as declining or agreeing to participate, is obviously unable to fully capture the whole nuances of family involvement in caregiving.

Lines 470-481: …it would be advantageous to integrate studies that encompass both objective and perceived measures of family involvement. While this study emphasizes the significance of specific decisions, such as the choice to participate or decline, in capturing facets of family involvement, it is crucial to acknowledge that these singular determinations may not fully encapsulate the intricate complexity and subtleties of this multifaceted phenomenon. By incorporating assessments that encompass objective indicators of involvement (e.g., extent of engagement, frequency of interactions) and subjective perceptions of involvement (e.g., perceived support, burden perception), researchers can attain a more holistic comprehension of the dynamic interplay between individuals with non-affective psychosis and their families. Such an approach would facilitate an enriched understanding of the multifaceted nature of caregiving and contribute to the development of comprehensive frameworks for intervention and support.

Additionally, new clarifications have been included in the abstract and the limitations section as suggested by the reviewer as follows:

Lines 25-26: a subdomain of family support, namely, family involvement (estimated through a proxy measure)…

Lines 424-425: However, at the same time, the use of a proxy approach to study FI may represent a limitation.

Comment #2: Terms “family involvement (FI)”, “family support (FS)” and “participant/non-participant families“ are used interchangeably through the manuscript (for example, in 2.6. Data analysis the authors wrote: “The influence of FS and sex was assessed …” (257), whereas there was no variable such as FS included in the results). Moreover, even the abbreviations were introduced for some of these terms, non-abbreviated forms “family involvement” and “family support” are used in addition to abbreviated ones, which may lead to impression that meaning of these terms are different in a particular context. Thus, revisions throughout the manuscript are further required.

Response #2: We thank the reviewer for the reviewer’s attention to detail. We acknowledge that during the extensive nomenclature revision undertaken in the first revision, some of the original FS terminologies inadvertently escaped our attention as was the case in line 257. We would like to inform that we have diligently rectified this oversight, modifying the terminology throughout the body of the manuscript in order to ensure consistency and clarity. Henceforth, we will be referring to formerly addressed as Family Involvement as estimated Family Involvement (eFI) except for the parts where we deliberately refer to the concept and not to our proxy measure.

Comment #3: Although the authors argue that the data are described enough to allow to reproduce the analyses, providing an appropriate description of the analyses in the Methods section is critical to also access whether the reported results are correctly analyzed and lead to conclusions described in the manuscript. Currently, this information is missing. As an option, the details could be included in supplementary materials, is the authors concern that “inclusion of additional statistical information in the main text may disrupt the flow of the content of the manuscript.”

Response #3: We thank the reviewer for the comment. Indeed, we agree that providing information that may replicate our findings is crucial and thank the reviewer for their understanding of our initial choice to prioritize the clarity and comprehensibility of the manuscript. We concur that incorporating supplementary material detailing the procedures employed in the analysis would be a fine solution that aligns with our objectives.

The initial phase involved examining essential assumptions in data analysis, including the normality of the distribution and the homogeneity of variances.

The Kolmogorov-Smirnov test is used to assess whether a sample of data follows a specific distribution, either normal or non-normal. It compares the empirical distribution of the data with the expected theoretical distribution and provides a measure of the discrepancy between the two. If the resulting p-value from the test is less than a predefined threshold (usually 0.05), the null hypothesis that the data follows the theoretical distribution is rejected. In summary, this test helps determine whether the data fits a normal distribution or not.

On the other hand, the Levene's test is used to assess the homogeneity of variances among groups or subgroups of data. It is particularly relevant in analysis of variance (ANOVA) and other methods that assume equality of variances among the compared groups. The Levene's test compares the variances across groups and provides a p-value that indicates whether there is statistical evidence of significant differences in variances. If the p-value is less than a predetermined threshold (usually 0.05), the null hypothesis of equal variances is rejected. In summary, this test allows evaluating whether the variances of the compared groups are statistically different or if the assumption of variance homogeneity can be upheld.

Consequently, variables were assigned to the classic parametric version of the ANCOVA, wherein age, years of education, and estimated premorbid intelligence quotient (epIQ) were considered as covariates. This assignment was made after verifying the assumptions of normality distribution and homogeneity of variances.

In cases where variables displayed non-normal distributions or heteroscedasticity, following previous literature, a rank transformation procedure, commonly known as the Quade's analysis [1-4], was employed as a non-parametric approach. Ranking the data considered the distributional shape and extreme values, minimizing the impact of outliers and deviations from normality. This transformation provided a more robust scale for analysis. Subsequently, ANCOVA analyses were conducted with the ranked data and the same covariates as in the parametric ANCOVA, enabling for covariance analysis while accommodating violations of the assumptions.

The rank transformation procedure employed is simply a means of facilitating the analysis by incorporating covariates in a originally non-parametric dataset. We would like to emphasize that the tables do not include the transformed data, but rather present the original data for reference. This decision was made considering that not all variables required the rank transformation procedure. By opting for a unified presentation of the original data, we aim to provide a more appropriate and comprehensive representation of the variables.

We understand the potential confusion this may have caused, and to provide further clarity, we have included a detailed description of the SPSS procedure for performing a rank transformation covariance analysis. The same description has been added in the supplementary materials section and highlighted in yellow (lines: 503-512):

Conducting ANCOVA Analysis with Rank Transformation in SPSS:

Step 1: Rank transformation: The dependent variable and covariates are ranked using the default settings in the SPSS RANK procedure. This process assigns ranks to the observed values, considering all cases and disregarding the grouping variable.

Step 2: Linear regression of ranks: A linear regression analysis is performed, with the ranks of the dependent variable as the outcome and the ranks of the covariates as predictors. The residuals (raw or unstandardized) are saved, ignoring the grouping factor.

Step 3: One-way ANOVA using residuals: An ANOVA is conducted using the saved residuals from the previous regression as the dependent variable, while the grouping variable serves as the factor. The resulting F statistic from this ANOVA represents the test statistic used in Quade's analysis.

An additional clarification in the data analysis section has also been added in the text as follows:

Lines 259-260: Prior to the analysis steps, the sample was sorted based on eFI and sex, as previously described on the sample selection process.

Comment #4: The authors wrote that the “normality” assumption was tested as “a standard procedure” [“As a standard procedure, normality and homogeneity assumptions of variance (respectively Kolmogorov-Smirnov and Levene tests) were conducted on each variable in order to select parametric or non-para- metric tests for further analysis.”(254-257)]. However, the normality assumption and parametric tests are not optimal or even inappropriate (unless justified by the authors) for several data types such as PAS, GDS, SAPS, SANS, etc., even, IQ and CED.

Response #4: We thank the reviewer for the comment. Please find a  response in response to comment #3.

Comment #5: P numbers reported in the tables and figures are confusing because they are not linked to descriptions of specific tests performed.

Response #5: We thank the reviewer for the comment. The data analysis section provides a thorough explanation of the rationale behind selecting specific tests for each variable, taking into consideration their individual characteristics (lines 255-265). The corresponding statistical measures (X2 and F levels) associated with the reported p-values in the tables are conveniently located in the respective statistics column for each variable.

The text has been revised to ensure that each table and figure is accompanied by a statement guiding readers to the relevant section where they can find the measurement methods employed for each specific case as follows:

All means of measurement can be found in the methods section.

In our view, including an additional column in the tables specifying the particular test (ANCOVA/QUADE, parametric/non-parametric) employed for each variable is unnecessary when replicating the analyses with a different sample. It is crucial for each new sample to independently assess the assumptions of normality and variance homogeneity to choose the best-fitting procedure for each variable, as we did in this study. Moreover, we believe that this presentation format offers a more concise and focused representation, highlighting the crucial information needed for replicating the analyses effectively.

Comment #6: Comparing non-statistically significant data does not provide any meaningful information (e.g., II > IV (p = 0.080, etc. in Table 2.)

Response #6: We thank the reviewer for the comment. We agree that conclusions should not be drawn after non-statistically significant results. However, we think statistical tendencies can provide interesting insights.

Although some p levels may not reach statistical significance according to the predefined thresholds set in our study, we believe it is worthwhile to highlight them due to their alignment with the proposed hypothesis and their contribution to a more comprehensive picture of the profile of certain subjects.

Specifically, we observed that women whose families did not participate in the family study exhibited comparatively poorer performances in working memory, motor dexterity, and global deficit score,  when compared to women whose families did participate. Furthermore, we observed smaller and non-significant differences among both groups of men, without any discernible statistical trends among these variables. Therefore, we consider these findings to be worth of mention within the potential disparity of our grouping variable impact among both sexes.

References

  1. Quade, D. Rank Analysis of Covariance. Journal of the American Statistical Association. 1967. 62:320, 1187-1200, DOI:10.1080/01621459.1967.10500925
  2. Olejnik, S.F.; Algina, J. Parametric ANCOVA and the rank transform ANCOVA when the data are conditionally non-normal and heteroscedastic. Journal of Educational Statistics. 1984, 9(2), 129–149. DOI:10.2307/1164717
  3. Olejnik, S.F.; Algina, J. An analysis of statistical power for parametric and rank transform ancova. Communications in statistics-theory and methods. 1987, 16(7), 1942-1949. DOI:10.1080/03610928708829481
  4. Conover, W. J.; Iman, R. L. Analysis of covariance using the rank transformation. Biometrics. 1982, 38(3), 715–724.